# Post-mating parental behavior trajectories differ across four species of deer mice

**Mehdi Khadraoui** [1,2¤], **Jennifer R. Merritt**[3], **Hopi E. Hoekstra** [1], **Andres Bendesky** [3]*

**1** Department of Organismic & Evolutionary Biology, Department of Molecular & Cellular Biology, Museum of Comparative Biology, Howard Hughes Medical Institute, Harvard University, Cambridge, Massachusetts, United States of America, **2** GELIFES - Groningen Institute for Evolutionary Life Sciences, Faculty of Science and Engineering, Groningen, The Netherlands, **3** Zuckerman Mind Brain Behavior Institute, Department of Ecology, Evolution & Environmental Biology, Columbia University, New York, New York, United States of America

¤ Current address: Communications Department, Research Institute of Molecular Pathology, Vienna, Austria
* a.bendesky@columbia.edu

**Data Availability Statement:** All relevant data are within the manuscript and its Supporting information files.

**Funding:** Consortium Scholarship of the University of Groningen to MK. Junior Fellowship of the

## Abstract

Among species, parental behaviors vary in their magnitude, onset relative to reproduction, and sexual dimorphism. In deer mice (genus *Peromyscus*), while most species are promiscuous with low paternal care, monogamy and biparental care have evolved at least twice under different ecological conditions. Here, in a common laboratory setting, we monitored parental behaviors of males and females of two promiscuous (eastern deer mouse *P. maniculatus* and white-footed mouse *P. leucopus*) and two monogamous (oldfield mouse *P. polionotus* and California mouse *P. californicus*) species from before mating to after giving birth. In the promiscuous species, females showed parental behaviors largely after parturition, while males showed little parental care. In contrast, both sexes of monogamous species performed parental behaviors. However, while oldfield mice began to display parental behaviors before mating, California mice showed robust parental care behaviors only postpartum. These different parental-care trajectories in the two monogamous species align with their socioecology. Oldfield mice have overlapping home ranges with relatives, so infants they encounter, even if not their own, are likely to be closely related. By contrast, California mice disperse longer distances into exclusive territories with possibly unrelated neighbors, decreasing the inclusive fitness benefits of caring for unfamiliar pups before parenthood. Together, we find that patterns of parental behaviors in *Peromyscus* are consistent with predictions from inclusive fitness theory.

## Introduction

Parental behaviors are widespread across the animal kingdom, yet they vary greatly in their type, intensity, and the degree of sexual dimorphism across species [1, 2]. In mammals, females are intimately associated with their progeny during gestation and lactation, and, in many species, continue caring for their young long after lactation ends. In contrast, lack of pregnancy and lactation allows males to seek additional mating partners instead of caring for their offspring. Consistent with these sexually dimorphic biological constraints on the commitments

Simons Society of Fellows (Simons Foundation) to JRM. HEH is an Investigator of the Howard Hughes Medical Institute. National Institutes of Health grants HD084732 and HD106241, Searle Scholarship, Klingenstein-Simons Fellowship in Neuroscience, and Sloan Fellow in Neuroscience to AB. The funders had no role in study design, data collection and analysis, decision to publish, or preparation of the manuscript.

**Competing interests:** The authors have declared that no competing interests exist.

to invest in offspring [3], in the majority of mammalian species, only mothers display robust parental care [4, 5]. However, in approximately 5% of mammalian species, both mothers and fathers display robust care for infants and nearly all of these species are monogamous [4]. Thus, in mammals, biparental care often co-evolves with monogamy [4–8].

In females, mating, pregnancy, and parturition are associated with profound neuroendocrine changes that induce parental care [9]. In males, parental behaviors can be set in motion by mating, shaped by exposure to the pregnant female, triggered by parturition of the partner, and later on maintained by pup cues [9–11]. The mechanisms for such behavioral transitions in males, however, are much less understood than in females. Reproductive events not only elicit the onset of parental behavior but, in some species, also inhibit infanticidal behavior: Males and females in many species do not show parental behaviors until they mate and may remain infanticidal until the birth of their own young [12, 13]. For example, in laboratory strains of *Mus musculus*, the suppression of infanticidal behaviors in adult males is triggered by mating and its onset is synchronized with the typical duration of pregnancy [11, 14, 15]. Although common, this pattern is not universal: the onset of parental behaviors (and the inhibition of infanticide) is subject to substantial variation among species [16]. For example, in species with alloparental care (i.e., care towards non-descendent young), such as the banded mongoose, spontaneous care for infants can occur in individuals that are reproductively suppressed and lack previous experience with infants ([17] and citations therein). Inclusive fitness theory suggests that alloparental care can evolve if the provider and the recipient of such care are genetically related enough that the indirect-fitness benefits outweigh the costs [18, 19]. However, the socioecological conditions that influence the onset of (allo)parental behavior are not well understood.

To understand the evolution of parental care and its onset, it is thus important to compare species that differ in both their mating system—monogamy versus promiscuity—as well as their socioecology. Here, we investigate the onset of parental behaviors in four species of deer mouse (genus *Peromyscus*; Fig 1). Monogamy and biparental care have evolved at least twice independently within this ancestrally promiscuous clade (Fig 1A; [20]). Previous work has

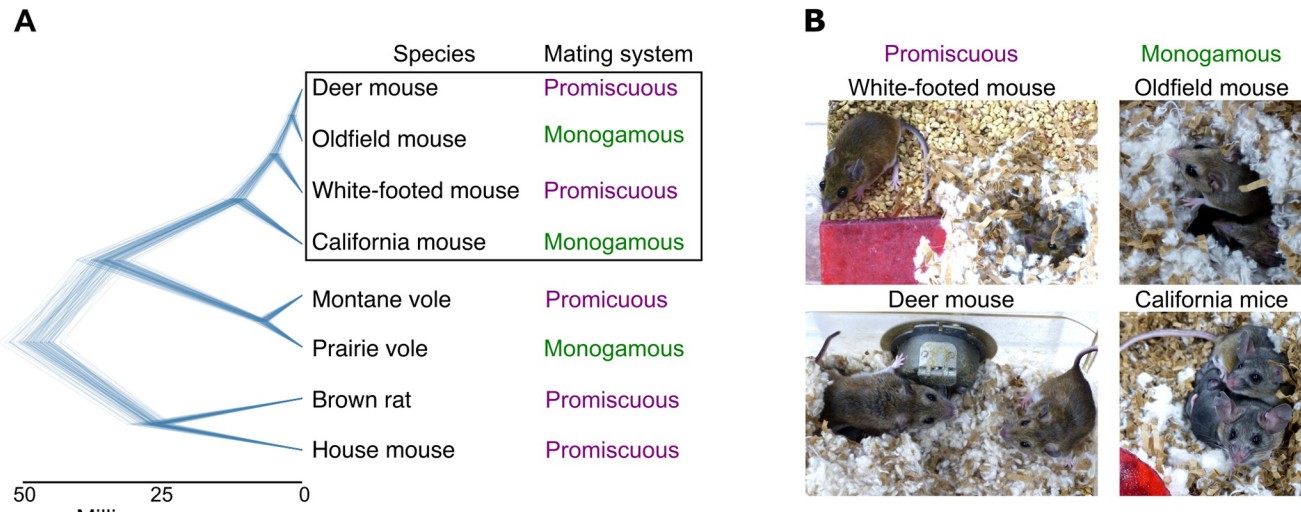

**Fig 1. Monogamy and biparental care have evolved at least twice independently in the genus *Peromyscus*. (A)** Phylogenetic relationships of the four species under study, relative to other rodent models. Each line denotes one of 100 trees sampled from a pseudo-posterior distribution of birth-death node-dated completed trees from http://vertlife.org/phylosubsets/ [31]. **(B)** Representative pictures of breeding pairs of each study species. Left: promiscuous species, typically with the female inside the nest and the male outside. Right: monogamous species, typically with both parents inside the nest.

shown that in both the eastern deer mouse *Peromyscus maniculatus bairdii* (hereafter deer mouse) and the white-footed mouse *P. leucopus*, males and females mate with multiple partners and only the mother maintains a close association with the offspring for several weeks postpartum [21–27]. By contrast, the oldfield mouse *P. polionotus* is monogamous and biparental [21, 26, 28]. It also displays cooperative breeding, in which juveniles often stay in their natal nest after weaning, contribute to care of subsequent litters, and disperse only short distances [23, 25, 26, 28]. Finally, the California mouse *P. californicus* is a territorially aggressive species in which males and females form lifelong pairs with exclusive territories far from their natal nest, and males show high levels of parental care [21, 25, 29, 30]. These four species therefore represent the two most abundant and widely distributed promiscuous species in the genus and the only two known monogamous species in the genus.

Here we followed mating pairs of these species throughout their first reproductive cycle and repeatedly measured four parental behaviors as well as recorded aggression towards pups before mating, during pregnancy, and after parturition. We then compared the resulting parental behavior trajectories across sexes and species to understand how they differ between these monogamous and promiscuous species.

## Materials and methods

### Animal husbandry

We focused on four species of *Peromyscus*: *P. maniculatus bairdii* (eastern deer mouse, strain BW), *P. leucopus* (white-footed mouse, strain LL), *P. polionotus subgriseus* (oldfield mouse, strain PO), and *P. californicus insignis* (California mouse, strain IS). These outbred colonies were originally established from animals of these strains obtained from the *Peromyscus* Genetic Stock Center, University of South Carolina, and then maintained at Harvard University. They were housed under 16 h light: 8 h dark at 22–23˚C. Smaller species (deer mouse, white-footed mouse, oldfield mouse) were housed in individually-ventilated standard Allentown cages (28.5 cm long x 19 cm wide x 16 cm high; Allentown, Allentown, NJ, USA) with ~1 cm deep Anderson's Bed-o-cob bedding (The Andersons, Inc., Maumee, OH, USA), while the larger species (California mouse) was housed in large Allentown cages (43 cm long x 22 cm wide x 27 cm high) with ~3 cm deep of bedding. Unmated animals were housed in social groups of two to five individuals of the same species and sex, were fed *ad libitum* with LabDiet Prolab Isopro RMH 3000 5P75 (LabDiet, St. Louis, MO, USA) and had unlimited access to water. Breeding pairs were housed together from the moment they were paired until the end of the experimental period. They were fed irradiated PicoLab Mouse Diet 20 5058 (LabDiet, St Louis, MO, USA) *ad libitum* and had free access to water. We provided all cages with 5 g/10 g (for small/large species, respectively) of compacted cotton (Nestlet, Ancare, Bellmore, NY, USA), ~10 g/30 g (small/large species, respectively) of paper fiber nesting material (Enviro-Dri, LBS Serving Biotechnology, UK), and a polycarbonate translucent red hut/tunnel (small/large species, respectively). All animal protocols were approved by the Harvard FAS Institutional Animal Care and Use Committee.

### Experimental timeframe and pairing

We started with 12 sexually naïve animals of each sex and strain. All experimental animals were tested for their parental behaviors (see **Parental behavior test** below) at four reproductive states: (1) as sexually naïve animals before pairing ("virgin"), (2) after copulation had happened or was expected to have happened (see below; "mated"), (3) when the female was expected to be in late pregnancy ("expecting"), and (4) after birth of the pups ("parent").

The median time between the first test and pairing was 4 days (standard error ± 1 day, no difference between species), and the median age of animals during the first test was 171 days (standard error ± 7 days, no difference between species). All animals were tested only once at each reproductive state because repeated exposure to pups can induce care for infants in some laboratory rodents such as rats [32]. If a pair's first litter did not survive more than three days, we tested the parents after their second litter was born, which occurred for two pairs of deer mice and one pair of white-footed mice. If a pair did not give birth to a surviving litter during the experimental period, we excluded it from the statistical analysis of parental behaviors. This resulted in sample sizes of 9 white-footed mouse pairs, 8 deer mouse pairs, 8 oldfield mouse pairs, and 7 California mouse pairs. For our measure of adult attacks on pups, we analyzed all 48 breeding pairs.

We monitored mating for three nights after pairing using the procedure described below (see **Mating monitoring** below). If we witnessed mating, the second series of tests was performed on days 1–2 post-mating. If we did not see evidence of mating, tests were performed after 5 nights (smaller species) or 6–7 nights (larger species), as this represents the approximate duration of an estrous cycle for these species, thus allowing the opportunity for mating during the receptive phase [33–35]. Of the 10 pairs in which we did not witness copulation, 5 gave birth to a surviving litter; the others either had no litter at all or dead litters (cause of death unknown). The third test for each animal was done on day 21 after mating or after the 5th night post-pairing if mating was not observed (smaller species), or on day 28 after mating or after the 6th night if mating was not observed (larger species). The final test was performed 2–6 days after the first surviving litter was born.

## Mating monitoring

We monitored mating using an in-house camera setup with a Raspberry Pi Zero and a Raspberry Pi NoIR camera (Raspberry Pi Foundation, UK) mounted on the cages and powered by a battery pack (see S1 File for a mounting tutorial). The Raspberry Pi was programmed to record 8.25 hours of video from 5 minutes before the light turned off in the room housing the mice to 10 minutes after the light turned on, as *Peromyscus* species are most active at night [36]. The key components used to characterize mating were: chasing (the male runs after the female), mounting, thrusting (the male's hips move back and forth), and post-coital licking (the male, and sometimes the female, lick their genitals after ejaculation). We considered mating to have occurred either if all the components were observed at least once, or if mounting and thrusting were observed repeatedly overnight.

## Parental behavior test

To characterize parental behavior, we followed a previously established assay [26], illustrated in Fig 2A. In brief, we tested animals during the light phase of the daily cycle (*Peromyscus* show similar levels of parental care in day and night [26]). Behavior testing ended at least 30 minutes before lights turned off. We first moved the animals with their home cage to a testing room, at least 5 minutes prior to the start of the assay. When we tested virgin mice, we transferred any existing nest, the red hut/tunnel, food hopper and all animals except the test animal from the home cage to a new one. When we tested breeding pairs, we carefully moved the male (and pups if present) out of the home cage to a new cage with the red hut/tunnel, nest, and part of his home bedding (to minimize the effect of this transfer on the observed behaviors), and left to habituate for 30 minutes while we first tested the female alone in her home cage. After the female test ended, we placed the red hut/tunnel and the nest (and pups if present) back with the female in the home cage, and tested the male's behaviors in the new cage,

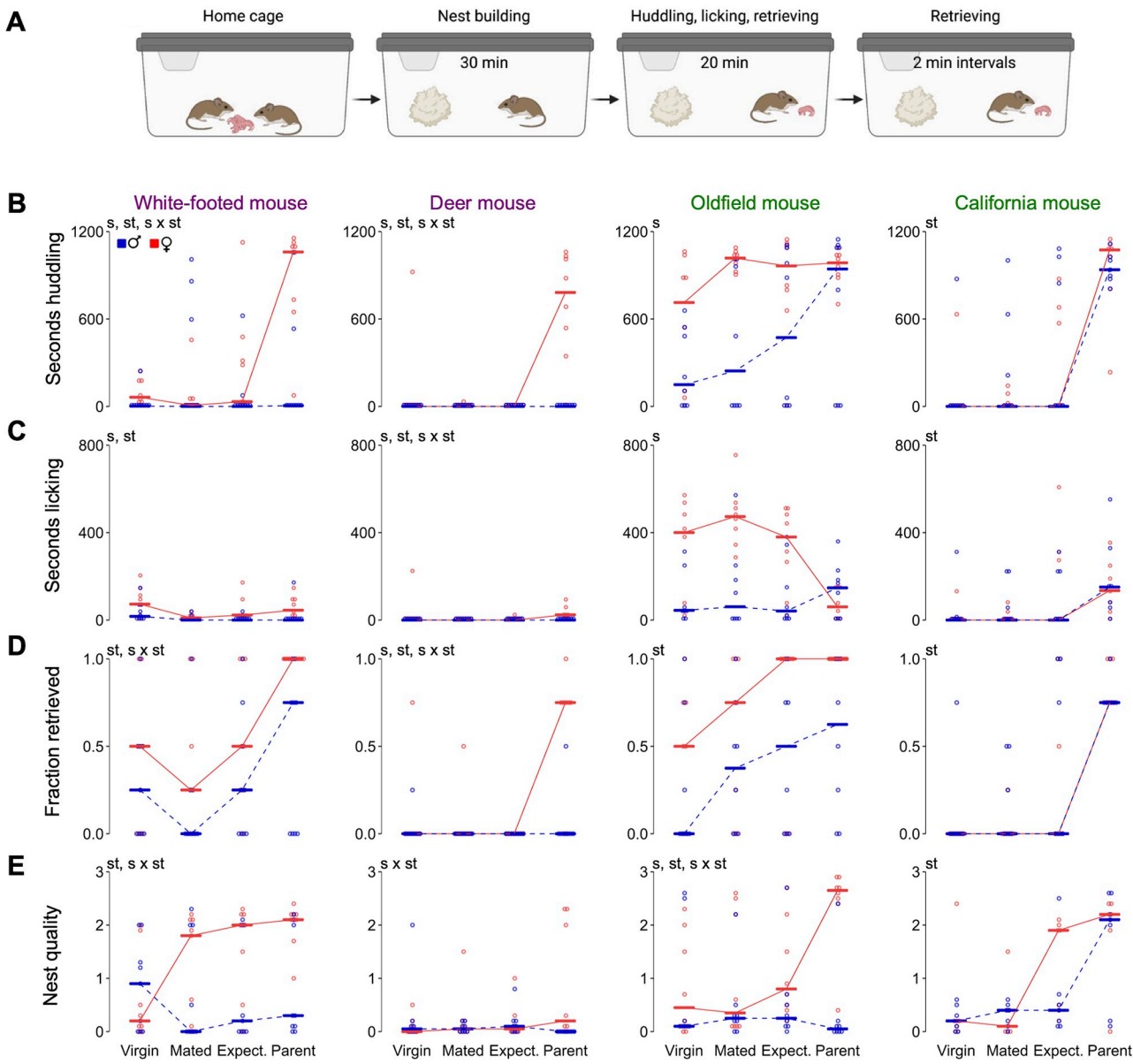

**Fig 2. Parental behaviors in four *Peromyscus* species and across four reproductive states. (A)** Schematic of behavioral assay showing the time and behaviors measured at each assay step. Male (blue) and female (red) trajectories as measured by **(B)** time spent huddling (seconds); **(C)** time spent licking pups (seconds); **(D)** fraction of pups retrieved to the nest; **(E)** nest quality score (from 0 to 4). Bars denote the median. st, main effect of reproductive state; s, main effect of sex; st × s, interaction between reproductive state and sex by linear mixed models ($P < 0.05$) see Materials and methods for details). Sample sizes (in pairs): white-footed mice, n = 9; deer mice, n = 8; oldfield mice, n = 8; California mice, n = 7.

after a further 30 minutes of habituation. This testing method minimizes stress to the male from transfer back to its cage again before testing. Animals were tested separately because in biparental species, partners can influence the degree of parental care the other parent provides [37, 38]. All animals were tested by the same experimenter.

To begin each behavioral assay, we gave the test individual ~0.625 g/2.5 g (small/large species, respectively) of compacted cotton (denoted as time 0). After 30 minutes, we placed an unfamiliar conspecific pup inside the cage, ~20 cm away from both the nest and the test individual. Pups were from other litters within the breeding colony and of a median age of 5 days

(standard error ±0.07 days). This approach allowed us to test in a consistent way the parental behaviors of animals before and after they have their own pups. *Peromyscus* parents provide the same level of parental care towards familiar and unfamiliar infants [26]. For the next 20 minutes, we recorded the total time the test animal huddled the pup (i.e., when the test animal covered at least 50% of the pup's body with its own body) and the time spent licking the pup. We also recorded when the pup was retrieved by the test mouse (i.e., when the test animal picked up the pup with its mouth and transported it). At minute 50, we scored the nest quality using the scoring system described below and removed the pup for 10 seconds, then added it back to the cage for 2 minutes (with the same placement strategy as described above) to determine whether the pup would be retrieved again. We repeated this procedure twice more for a total of 4 pup retrieval trials, and then ended the trial.

The observer scored each assay ~100–150 cm away from the cage by observing the cage directly while video-recording the test from the opposite side of the cage to allow for later verification if the behavior was not fully clear from the perspective of the observer. When possible, the same pup was not used for two consecutive tests. When the availability of pups was limited, we left the pup for 10 minutes in an incubator at 37°C between two tests. At the end of the trial, we returned the pup to its parents and ensured that it was cared for (licked and/or retrieved by a parent). In the event of a pup attack (the test individual bites the pup), we immediately stopped the assay, removed the pup, and did not reuse that pup in subsequent trials. All attacks were confirmed by the presence of bite marks or blood on the pup. The pup was returned to its parents and we observed the parents' behavior to ensure the pup was taken care of (licked and huddled).

## Nest quality score

To quantify nest-building behavior, we used an established scoring system [26]: 0 = nesting material is untouched; 1 = all of the nesting material is shredded and scattered; 2 = nesting material is shredded and gathered in a platform; 3 = nesting material is shredded, gathered in a nest and forms walls that are as high as the test animal; 4 = nest covers the entire animal including a complete roof. Nests that were between two discrete categories were given an intermediate score. All nests were scored by a single researcher for consistency.

## Data analysis

All the raw data is provided in S2 File. Statistical analyses were performed in the R language and environment [39]. We tested for normality using the Shapiro-Wilk test and visually evaluated the distribution of the data using qq-plots. Because Bendesky et al. [26] found low genetic correlations between each parental behavior in two of the species we tested, we conducted separate analyses of each behavior: huddling, licking, pup retrieval, and nest quality. None of the behaviors were normally distributed, so we log transformed the data for analysis, after adding a small constant value to avoid taking the log of 0. Each parental behavior was then analyzed using linear mixed-effects models with a Gaussian distribution using the package *lme4* (v1.1–27.1; [40]) including the main effects of group (mating system or species), sex, and reproductive state (virgin, mated, expecting, parent), all 2- and 3-way interactions between these variables, and the random effect of individual. We first ran models with the fixed effect of mating system, then performed separate analyses within monogamous and promiscuous groups to test for an effect of species.

We tested for 2-way interactions as follows: (1) significant reproductive state × sex, to test for an effect of reproductive state and sex within group (mating system or species), (2) significant group × state effect, to test for the effect of reproductive state and group within sex, and

(3) significant group × sex, to test for the effect of group and sex within reproductive state, followed by pair-wise comparisons. Wald chi-squared tests were used to generate analysis-of-deviance summary tables using the *car* package (3.0–12; [41]). The α level was set at $P \leq 0.05$. Infanticidal behaviors occurred too infrequently for statistical analysis.

## Results

### Trajectories of parental behavior in promiscuous species

In promiscuous deer mice and white-footed mice, both males and females displayed very little to no huddling and licking behaviors prior to the birth of pups (Fig 2B and 2C). When pups were born, only mothers significantly huddled and licked the pups (reproductive state × sex, $P < 0.001$ in each species for each behavior). Deer mouse males retrieved a median of zero pups across reproductive states and females retrieved at high rates only after the birth of their pups (female reproductive state $P < 0.001$; Fig 2D). Unexpectedly, pup retrieval in white-footed mice of both sexes showed a J-shape, with moderate levels before mating followed by a decrease after mating and highest levels after giving birth to pups (reproductive state $P < 0.001$; Fig 2D). Males of both promiscuous species were poor nest-builders (Fig 2E). Deer mouse females also barely built a nest at any reproductive state, whereas white-footed mouse females increased their nest building after mating (white-footed female reproductive state $P = 0.003$) (Fig 2E). These behavioral patterns are overall consistent with a model of maternally biased investment in offspring predicted for promiscuous species.

### Trajectories of parental behavior in monogamous species

In California mice, huddling increased from a median of zero seconds in both sexes before parturition to over 1,000 seconds after birth. In this species, licking also increased from a median of 0 seconds before birth to ~150 seconds after birth (Fig 2C). Unexpectedly, the increase in huddling in oldfield mice was not a sudden change after birth of their litter, but rather a gradual increase after mating, especially in males. Before becoming parents, males huddled with pups less than females (main effect of sex $P = 0.01$), but after the birth of their pups, huddling became indistinguishable between the sexes ($P = 0.17$; Fig 2B). Licking behavior was different in oldfield mouse females and males ($P = 0.001$; Fig 2C): females decreased pup licking from a high level of ~400 seconds before birth to ~50 seconds after birth, while males showed the opposite pattern from ~20 seconds of licking before birth to ~150 seconds after birth (more similar to the transition observed in California mice). The stage at which huddling and licking behavior emerged differed significantly by species (species × reproductive state $P < 0.001$ for each behavior; Fig 2B and 2C), indicating that these two monogamous species follow two distinct trajectories towards biparental care.

The trajectories of pup retrieval closely mimicked those of huddling, without overall sex differences (sex $P = 0.26$; reproductive state × sex $P = 0.09$; Fig 2D): In oldfield mice, the increase in retrieval in both sexes was gradual following mating, peaking after the birth of pups (reproductive state $P < 0.001$; sex $P = 0.10$; sex × reproductive state $P = 0.65$; Fig 2D), while in California mice both males and females had an abrupt transition from retrieving no pups before giving birth to retrieving a median of 80% of pups after birth. We did not observe any post-parturition differences between males and females in this species in huddling (sex $P = 0.70$; sex × reproductive state $P = 0.96$), licking (sex $P = 0.72$; sex × reproductive state $P = 0.85$) or pup retrieval (sex $P = 0.54$; sex × reproductive state $P = 0.68$). Thus, parental investment in California mice is not exhibited until the birth of pups and does not differ between the sexes.

In oldfield mice, there was a robust increase in nest building in females after giving birth (reproductive state $P = 0.006$), which contrasted with the more gradual changes in their other

parental behaviors, while males had low nest building across all reproductive states (male reproductive state $P = 0.193$; sex × state $P = 0.001$; Fig 2E). In California mice, males and females built more elaborate nests as parents than before mating (reproductive state $P < 0.001$; Fig 2E).

### Trajectories of infanticidal behavior

All four species showed some aggression towards pups before the birth of their own pups, but never after they had a litter (Fig 3). Adult attacks on pups were recorded in up to 25% of male and female deer mice and white-footed mice before the birth of their pups. Of all four species, California mice had the highest infanticide rate, with 50% of females and 33% of males attacking pups before the birth of their own young (Fig 3). By comparison, in oldfield mice, only 17% of females and 8% of males attacked pups before parturition.

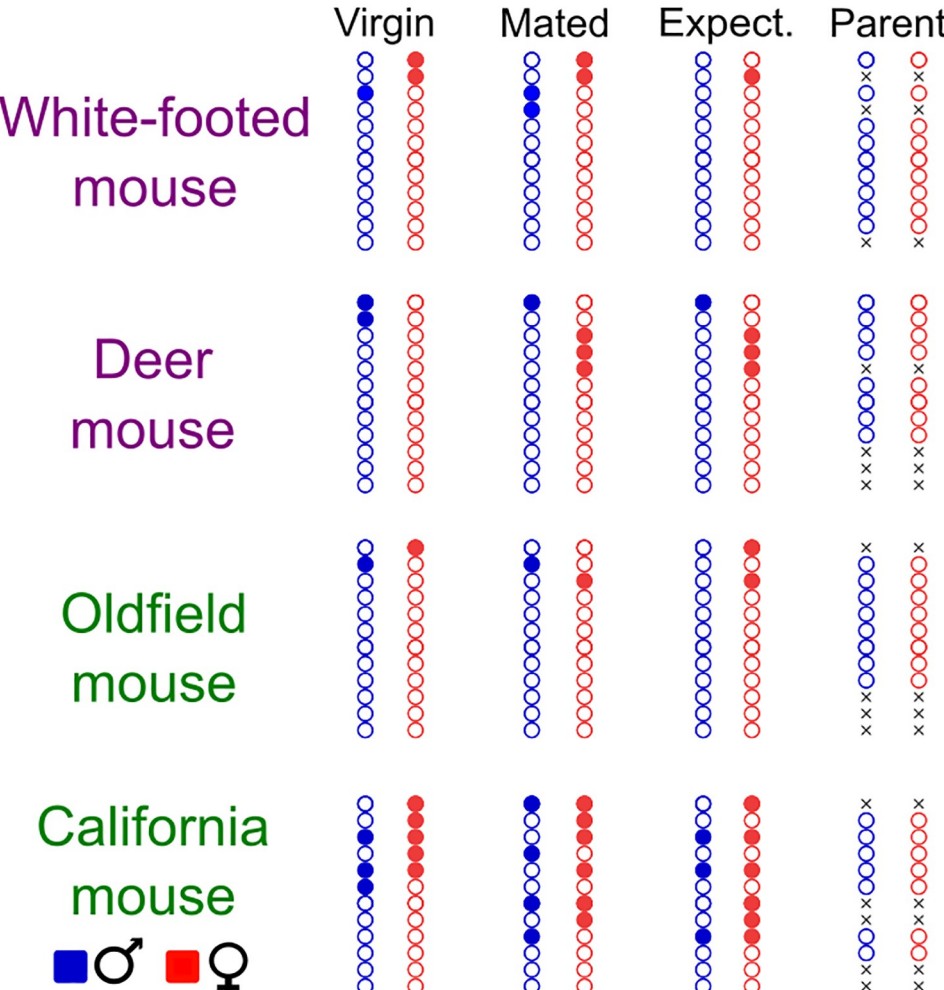

**Fig 3. Infanticidal behaviors in four *Peromyscus* species and across four reproductive states.** Circles denote behavior of male (blue) and female (red) mice (filled: attacked the pup; empty: no attack). Pairs that did not produce a litter that survived to at least the third day were not tested as parents (denoted by crosses). The behavior of individual mice across reproductive states can be tracked horizontally. All experimental pairs are included as virgin, mated and expecting mice (n = 12 pairs for each species).

## Discussion

In mammals, parental care strategies can vary tremendously among species, and this variation has been most tightly correlated with differences in mating system [4]. For example, in species in which females usually mate with a single male (genetic monogamy), a male has high certainty of paternity, incentivizing his care of the young. In addition, ecological factors that affect the relatedness of individuals across space can shape the indirect fitness animals derive by caring for young that are genetically related but not their own [42]. Here, we examined parental care before and after critical reproductive events across four closely related deer mouse species that have either a monogamous or promiscuous mating system as well as vary in typical dispersal distance and social structure.

Overall, promiscuous deer mice showed the least parental behavior of the four species under study. Consistent with Bendesky et al. [26], in this species only females spent a substantial amount of time engaging in parental behavior after parturition, while males rarely interacted with pups either before or after mating. In the wild, the home ranges of deer mouse males usually overlap with those of multiple females [33, 43], and females often have litters sired by multiple males [43]. Given the low certainty in paternity, coupled to the opportunities for additional matings, males may benefit from seeking those new mating opportunities rather than investing in paternal care of their mates' pups, which may not be their own.

In the second promiscuous species, white-footed mice, fathers also spent little time licking and huddling their pups. However, the nest building was higher in both sexes of white-footed mice than in deer mice. The relatively high levels of pup retrieval that some white-footed mouse males displayed were unexpected, since in two studies conducted in a natural setting, male white-footed mice provided no care to their litter and were aggressively excluded from the nest by their mate ([22, 24] and citations therein). This discrepancy might reflect some behavioral plasticity related to testing conditions, as male white-footed mice have shown parental care behaviors in other laboratory studies (e.g. [44]). A similar phenomenon has been described in the promiscuous montane vole (*Microtus montanus*), in which males provide no care to their litter in the wild, whereas they often contribute to nest building and to pup retrieval in captivity [45], suggesting that ecological constraints on these species can shape the expression of parental behavior. Alternatively, genetic variation among white-footed mouse populations could contribute to different behavioral outcomes. Indeed, variation in parental and mating systems has been observed in wild white-footed mice ([33] and citations therein), in which different populations and subspecies display various levels of biparental care. Thus, our results support the idea that white-footed mouse males can show paternal behaviors under some circumstances.

By contrast, monogamous oldfield mice showed relatively high levels of parental care at all reproductive states, especially by females, uncovering high levels of pre-mating alloparental care that were not known in this species. These results are consistent with evidence that oldfield mice are cooperative breeders in the wild, where subadults often remain in their natal burrow alongside their parents and the subsequent litter [28, 46]. Through the rearing of a younger litter of siblings, juveniles (especially females) become more successful parents while increasing indirect fitness benefits [46]. Since mice in our experiments were not exposed to younger siblings in their natal cage, our results suggest that this strong tendency for alloparental care may have a genetic basis.

An ecological link to the oldfield mice care for pups that are not their own, before parenthood, may be the limited dispersal and lack of territoriality of this species in the wild. Two studies on the Alabama beach mouse (*P. polionotus ammobates*) described the dispersal of subadults as short movements away from their natal home range, on average less than 200 meters

[47, 48]. In addition, neighboring home ranges are non-exclusive and therefore overlap with those of close relatives [47, 48]. Thus, mice might increase their inclusive fitness by caring for unfamiliar pups they encounter, as they are likely to be close relatives. A similar scenario occurs in the prairie vole (*Microtus ochrogaster*): In this species, dispersal is limited, and unmated individuals display spontaneous alloparental behaviors regardless of prior experience with pups [49, 50]. As in oldfield mice, philopatry provides indirect fitness benefits to sub-adults [28, 51, 52].

In contrast to oldfield mice, California mice—the other monogamous species we studied—were rarely parental before birthing young and were rather aggressive towards pups before they had their own litter. These results are consistent with the territoriality and spacing patterns of this species in the wild. California mice are an aggressively territorial species, and the adjacent home ranges of mating pairs are strictly exclusive, ranging from a few hundreds to several thousands of square meters [33, 53]. Females are particularly territorial and typically avoid breeding within their natal home range [54]. These characteristics make it likely that unfamiliar pups found within the home range of an adult are that of unrelated intruders. As a result, an unmated adult California mouse would not benefit from caring or tolerating unfamiliar pups they encounter after they leave their natal nest. This idea is supported by the relatively high rate of infanticidal attacks observed here and reported previously [10].

Part of the behavioral differences between groups may result from variation in stress tolerance among sexes, species, and reproductive states. For instance, in California mouse fathers, replacing a cage's lid can inhibit paternal behaviors, even after 10 minutes of habituation [55, but see 56 where virgins do not show that response after 15 minutes of habituation]. We expect, however, that our 60 minutes of habituation were enough to mitigate the effects of stress. Sex differences may also have been influenced by our testing protocol, which was designed to minimize animal stress: While females were tested in their home cage, males were habituated for 60 minutes in a new cage and then tested in that cage.

Together, our comparative within-subjects study of parental behaviors across *Peromyscus* species under common, controlled laboratory conditions confirm that promiscuous species take little to no care of unfamiliar pups until their own litter is born, and only mothers display robust care for infants. In addition, we show that two species, which diverged approximately 10 million years ago and evolved monogamy independently, strongly differ in their trajectories towards high maternal and paternal care. This suggests that the genes and molecular mechanisms underlying the onset of parental care behaviors might differ between these species and that there is more than one way to make a dedicated father, even within a genus.

## Supporting information

**S1 File. Mating monitoring with the Raspberry Pi Zero.** This tutorial describes how to construct a home-made set up to record up to 9 hours of video of a standard Allentown mouse or rat cage in the dark (includes S1–S6 Figs).
(DOCX)

**S2 File. Raw data.** This excel file contains all the raw behavioral data collected for this study and used for making the figures and statistical analyses.
(XLSX)

**S1 Fig. Individual variation in parental behaviors across four reproductive states.** Male (blue) and female (red) trajectories as measured by (A) Time spent huddling (seconds); (B) time spent licking pups (seconds); (C) fraction of pups retrieved to the nest; (D) nest quality score (from 0 to 3). Individuals have unique line types. st, main effect of reproductive state; s,

main effect of sex; st × s, interaction between reproductive state and sex by linear mixed models ($P < 0.05$; see Fig 2 and Materials and methods for additional details). Sample sizes (in pairs): white-footed mouse, n = 9; deer mouse, n = 8; oldfield mouse, n = 8; California mouse, n = 7.
(TIFF)

## Acknowledgments

We thank Kyle Turner, Edward Soucy, and Adam Bercu for technical help. Christoph Gebhardt provided comments on the manuscript. MK was part of the Erasmus Mundus Masters Programme in Evolutionary Biology (MEME).

## Author Contributions

**Conceptualization:** Mehdi Khadraoui, Hopi E. Hoekstra, Andres Bendesky.

**Data curation:** Mehdi Khadraoui.

**Formal analysis:** Mehdi Khadraoui, Jennifer R. Merritt, Andres Bendesky.

**Funding acquisition:** Hopi E. Hoekstra, Andres Bendesky.

**Investigation:** Mehdi Khadraoui, Andres Bendesky.

**Project administration:** Hopi E. Hoekstra, Andres Bendesky.

**Supervision:** Hopi E. Hoekstra, Andres Bendesky.

**Visualization:** Mehdi Khadraoui, Jennifer R. Merritt, Andres Bendesky.

**Writing – original draft:** Mehdi Khadraoui, Andres Bendesky.

**Writing – review & editing:** Mehdi Khadraoui, Jennifer R. Merritt, Hopi E. Hoekstra, Andres Bendesky.

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
