## [Decision Letter · Decision Letter 0]

1 Sep 2022

PONE-D-22-22997Post-mating parental behavior trajectories differ across four species of deer micePLOS ONE

Dear Dr. Bendesky,

Thank you for submitting your manuscript to PLOS ONE. After careful consideration, we feel that it has merit but does not fully meet PLOS ONE’s publication criteria as it currently stands. While the two reviewers are positive about your study, they clearly outline their concerns with the manuscript. If you feel that you can fully address these concerns, we invite you to submit a revised version of the manuscript that addresses the points raised during the review process.

We look forward to receiving your revised manuscript.

Kind regards,

Andrey E Ryabinin, Ph.D.

Academic Editor

PLOS ONE

Journal Requirements:

Reviewers' comments:

Reviewer's Responses to Questions

**Comments to the Author**

1. Is the manuscript technically sound, and do the data support the conclusions?

Reviewer #1: Partly

Reviewer #2: Yes

2. Has the statistical analysis been performed appropriately and rigorously? 

Reviewer #1: Yes

Reviewer #2: Yes

3. Have the authors made all data underlying the findings in their manuscript fully available?

Reviewer #1: Yes

Reviewer #2: Yes

4. Is the manuscript presented in an intelligible fashion and written in standard English?

Reviewer #1: Yes

Reviewer #2: Yes

5. Review Comments to the Author

Reviewer #1: In this interesting manuscript the authors conduct detailed analyses of parental behavior in four species of Peromyscus that differ in their social organizations. The authors show that in two species (P. polionotus and P. californicus) that males engage more parental behavior than males in P. leucopus or P. maniculatus. Although parental behavior in these species have been studied previously, the paper has several strengths. The same procedure is used for all four species, four time points are observed, mating is carefully observed, and the authors include a diversity of behavior variables. There is one important weakness in that males and females are not tested in the same way even though they are directly compared. Although not ideal, I believe the authors could address this in the discussion and the abstract to make a good contribution to the literature.

The main issue is that behavior testing protocol is a little unconventional. The females are tested in the home cage first while the males are moved to a new cage (with pups if they are present). Then, the male is tested in a new cage. Even though the males have 60 minutes of habituation in the new cage, the transfer to the new cage is stressful. Thus as written, the females are tested in the home cage and the males in a novel environment.

The authors need to consider the potential impact of stress via the handling of their procedure may have on behavior. Simply removing a wire cage lid from a cage can have a strong inhibitory effect on parental behavior in P. californicus (Kowalczyk et al. 2018). It’s possible the handling may have contributed to the lack of parental care in virgin P. californicus in contrast to work from the Saltzman lab which has observed more paternal behavior in virgin P. californicus (eg Nguyen et al 2020) (although Gubernick reported more infanticide in virgin males). Physiological stress responses and behavioral responses to novelty have been studied in some of these species, but I’m not aware of all four being directly compared to each other. The authors need to consider the possibility that the reason why P. polionotus show male parental behavior sooner is that those males are less sensitive to novel environments than P. californicus. This should be addressed in the discussion and the abstract.

It looks like some of the references are mixed up. On line 40 the references for the mice being nocturnal and mating at night are 28 and 33 which are about social organization of polionotus and californicus.

The quality of the supplementary figures is not very good. Even when I zoomed in the graphs were very pixelated. I couldn’t tell if this was a problem with the journal website or the original figures.

Reviewer #2: In this manuscript, the authors characterized parental behavior in 4 species of mice from pre-mating to post-parturition. The species of mice they studied varied by mating system (monogamous vs. promiscuous) as well as species-typical dispersal patterns and territory size. Their study found results for female parental behavior that supports previous findings in the literature, yet they also provide novel data about male parental trajectories in these species. The characterization of paternal care patterns in these species provides a novel and valuable contribution to the field.

The manuscript was well-written and appropriately succinct. I have only 2 minor comments.

Methods

- Was cause of death for the deceased 1st litters examined? It just seems like a mark against parents if they killed their pups… but if the cause of death was dehydration or malnutrition, then it would suggest a health issue such that perhaps the mother wasn’t producing milk. The latter cause of death wouldn’t be concerning for this dataset, but the former (i.e., killing pups) may have some implications for species-typical parenting.

Discussion

- Does the habitat differ between white-footed and deer mice? Can the nest building skills be attributed to what these mice evolved to use for nesting material in the wild (i.e., does one species actually build nests using grass/pine needles/etc. and another species just burrows in dirt or lives in hollowed out parts of logs, never really building nests)?

6. PLOS authors have the option to publish the peer review history of their article (what does this mean?). If published, this will include your full peer review and any attached files.

Reviewer #1: No

Reviewer #2: No

---

## [Author Response · Author response to Decision Letter 0]

18 Sep 2022

Below you will find detailed answers (in red) to the Journal requirements and to the reviewers’ comments below.

Journal requirements

We revised our manuscript to meet the requirements from these two links. Our first page now matches the required format, and the names of our supplemental files have been changed as well.

We provide correct grant numbers in the ‘Funding Information’ section. 

IACUC approval was already included in the Methods section.

We have now reviewed the list of references to ensure it is complete and correct. 

Reviewer 1

In this interesting manuscript the authors conduct detailed analyses of parental behavior in four species of Peromyscus that differ in their social organizations. The authors show that in two species (P. polionotus and P. californicus) that males engage more parental behavior than males in P. leucopus or P. maniculatus. Although parental behavior in these species have been studied previously, the paper has several strengths. The same procedure is used for all four species, four time points are observed, mating is carefully observed, and the authors include a diversity of behavior variables. There is one important weakness in that males and females are not tested in the same way even though they are directly compared. Although not ideal, I believe the authors could address this in the discussion and the abstract to make a good contribution to the literature.

The main issue is that behavior testing protocol is a little unconventional. The females are tested in the home cage first while the males are moved to a new cage (with pups if they are present). Then, the male is tested in a new cage. Even though the males have 60 minutes of habituation in the new cage, the transfer to the new cage is stressful. Thus as written, the females are tested in the home cage and the males in a novel environment.

We thank reviewer 1 for this relevant comment. We have now addressed this weakness in the Discussion section (lines 405-408). We acknowledge that our testing protocol may have influenced sex differences in parental behavior. We designed our testing protocol in this manner to reduce the stress on females and their litters that would have been incurred if we had to individually test the male in his home cage in the absence of the female. We also believe that the long habituation phase of 60 minutes mitigated possible effects on male behaviors.

The authors need to consider the potential impact of stress via the handling of their procedure may have on behavior. Simply removing a wire cage lid from a cage can have a strong inhibitory effect on parental behavior in P. californicus (Kowalczyk et al. 2018). It’s possible the handling may have contributed to the lack of parental care in virgin P. californicus in contrast to work from the Saltzman lab which has observed more paternal behavior in virgin P. californicus (eg Nguyen et al 2020) (although Gubernick reported more infanticide in virgin males). Physiological stress responses and behavioral responses to novelty have been studied in some of these species, but I’m not aware of all four being directly compared to each other. The authors need to consider the possibility that the reason why P. polionotus show male parental behavior sooner is that those males are less sensitive to novel environments than P. californicus. This should be addressed in the discussion and the abstract.

Reviewer 1 highlights inconsistencies in the literature about the effect of stress via handling on behavior of different species or groups. This is an important point, and we have now addressed this possible effect in our discussion (lines 401-405). We have also added the references mentioned by reviewer 1 to show that short habituation phases after cage manipulations can have an effect on behavior. We expect that 60 minutes of habituation (4 to 6 times longer than the references mentioned by the reviewer) were enough to mitigate the effects of stress.

It looks like some of the references are mixed up. On line 40 the references for the mice being nocturnal and mating at night are 28 and 33 which are about social organization of polionotus and californicus.

We have now revised our references and ensured that they corresponded to the points made throughout the paper. The particular references highlighted by reviewer 1 were changed to a new reference.

The quality of the supplementary figures is not very good. Even when I zoomed in the graphs were very pixelated. I couldn’t tell if this was a problem with the journal website or the original figures.

We have checked our supplementary figure (S7) and made sure our file was of high resolution. We’ve noticed Plos PDFs for reviewers often have a low resolution figure with a link to the high resolution version.

Reviewer 2

In this manuscript, the authors characterized parental behavior in 4 species of mice from pre-mating to post-parturition. The species of mice they studied varied by mating system (monogamous vs. promiscuous) as well as species-typical dispersal patterns and territory size. Their study found results for female parental behavior that supports previous findings in the literature, yet they also provide novel data about male parental trajectories in these species. The characterization of paternal care patterns in these species provides a novel and valuable contribution to the field.

The manuscript was well-written and appropriately succinct. I have only 2 minor comments.

Methods

Was cause of death for the deceased 1st litters examined? It just seems like a mark against parents if they killed their pups… but if the cause of death was dehydration or malnutrition, then it would suggest a health issue such that perhaps the mother wasn’t producing milk. The latter cause of death wouldn’t be concerning for this dataset, but the former (i.e., killing pups) may have some implications for species-typical parenting.

We have not examined the cause of death of lost litters, since this is very challenging. Dead pups can die from infanticide or be injured or eaten after death from other causes. In some cases, we could not be sure there had been a litter at all, as young parents can eat their young very soon after birth and leave no trace. This point is now addressed under Experimental timeframe and pairing in the Materials and Methods section (lines 165).

Discussion

Does the habitat differ between white-footed and deer mice? Can the nest building skills be attributed to what these mice evolved to use for nesting material in the wild (i.e., does one species actually build nests using grass/pine needles/etc. and another species just burrows in dirt or lives in hollowed out parts of logs, never really building nests)?

The habitat preferences of P. leucopus and P. maniculatus bairdii partially overlap. While interspecific differences in the habitat adaptations and nesting-material use of our species may have had an effect on our findings, we preferred to keep speculation at a minimum in the discussion.

---

## [Decision Letter · Decision Letter 1]

28 Sep 2022

Post-mating parental behavior trajectories differ across four species of deer mice

PONE-D-22-22997R1

Dear Dr. Bendesky,

We’re pleased to inform you that your manuscript has been judged scientifically suitable for publication and will be formally accepted for publication once it meets all outstanding technical requirements.

Kind regards,

Andrey E Ryabinin, Ph.D.

Academic Editor

PLOS ONE

Additional Editor Comments (optional):

Reviewers' comments:

Reviewer's Responses to Questions

**Comments to the Author**

1. If the authors have adequately addressed your comments raised in a previous round of review and you feel that this manuscript is now acceptable for publication, you may indicate that here to bypass the “Comments to the Author” section, enter your conflict of interest statement in the “Confidential to Editor” section, and submit your "Accept" recommendation.

Reviewer #1: All comments have been addressed

Reviewer #2: All comments have been addressed

2. Is the manuscript technically sound, and do the data support the conclusions?

Reviewer #1: Yes

Reviewer #2: Yes

3. Has the statistical analysis been performed appropriately and rigorously? 

Reviewer #1: Yes

Reviewer #2: Yes

4. Have the authors made all data underlying the findings in their manuscript fully available?

Reviewer #1: Yes

Reviewer #2: (No Response)

5. Is the manuscript presented in an intelligible fashion and written in standard English?

Reviewer #1: Yes

Reviewer #2: Yes

6. Review Comments to the Author

Reviewer #1: nice revision, this is a great paper. It's strange that this journal requires 100 character response when you are recommending acceptance.

Reviewer #2: (No Response)

7. PLOS authors have the option to publish the peer review history of their article (what does this mean?). If published, this will include your full peer review and any attached files.

Reviewer #1: No

Reviewer #2: No

---

## [Editor Report · Acceptance letter]

7 Oct 2022

PONE-D-22-22997R1 

Post-mating parental behavior trajectories differ across four species of deer mice 

Dear Dr. Bendesky:

I'm pleased to inform you that your manuscript has been deemed suitable for publication in PLOS ONE. Congratulations! Your manuscript is now with our production department. 

Kind regards, 

on behalf of

Dr. Andrey E Ryabinin 

Academic Editor

PLOS ONE